# Unsupervised Representation Learning Facilitates Human-like Spatial Reasoning

**Kaushik Lakshminarasimhan**
Center for Theoretical Neuroscience
Columbia University
New York, NY 10027
jl5649@columbia.edu

**Colin Conwell**
Department of Psychology
Harvard University
Cambridge, MA 02139
conwell@g.harvard.edu

## Abstract

When judging the sameness of three-dimensional (3D) objects that differ by a rotation, response time typically increases with the angle of rotation. This increase is usually taken as evidence for mental rotation, but the extent to which low-level perceptual mechanisms contribute to this phenomenon is unclear. To investigate this, we built a neural model that breaks down this computation into two stages: a fast feedforward stage that extracts low-dimensional latent representations of the objects being compared, and a slow recurrent processing stage that compares those representations to arrive at a decision by accumulating evidence at a rate that is proportional to the proximity of the representations. We found that representation of 3D objects learned by a generic autoencoder was sufficient to emulate human response times using this model. We conclude that perceptual representations may play a key role in limiting the speed of spatial reasoning. We discuss our findings in the context of the mental rotation hypothesis and identify additional, as yet unverified representational constraints that must be satisfied by neural systems that perform mental rotation.

## 1 Introduction

William James, the father of American psychology, called the sense of sameness *the very keel and backbone of our thinking*. Unlike object recognition which happens within milliseconds of visual stimulation [1], reasoning about the sameness of two objects is, in general, thought to be an effortful cognitive process requiring deliberation. But why? In a classic experiment, Shepard and Metzler [2] asked participants whether images of three-dimensional (3D) objects that differed by a rotation, depicted the same object or two different objects. They found that human response times increased with the angle of rotation. This famous result is generally taken to imply that sameness judgement in spatial reasoning tasks entails "mental rotation", a type of mental simulation in which one of the objects is rotated in the mind's eye until it is aligned with the other [3].

While the above account seems reasonable at the outset, it has precipitated an intense debate that continues to this day [4, 5, 6]. First, the rotation hypothesis does not specify a neural mechanism by which the decision is made. Consequently, there is no physiological evidence for mental rotation till date. Second, it ignores the nature of the visual representations that underlie spatial reasoning. It has recently come to light that feedforward neural network models can achieve human-like performance in physical prediction tasks that were previously thought to require mental simulation [7, 8]. Therefore, it is conceivable that visual representations learned by feedforward networks may also be able to support spatial reasoning without mental rotation. Specifically, we hypothesized that image pairs that differ by larger rotations would be further apart in the neural representational space.

3rd Workshop on Shared Visual Representations in Human and Machine Intelligence (SVRHM 2021) of the Neural Information Processing Systems (NeurIPS) conference, Virtual.

To test this, we built a neural network model comprised of an autoencoder that extracts low-dimensional latent representation of synthetic images of 3D objects, and a recurrent neural network that operates on the latent embeddings of pairs of images to make decisions. We show that the representation learned by the autoencoder is informative about the 3D orientation of objects, and can account for the rotation-dependent response times without any dedicated machinery for mental simulation.

## 2 Methods

We ran experiments where we generated synthetic stimuli, trained neural network models using those data, analyzed the representation learned by the model and evaluated the model using separate test stimuli. Below, we describe each of these steps in detail.

### 2.1 Stimulus

We generated a dataset of 25,000 training images and 200 test images using Blender 3D rendering engine (Fig. S1 in Appendix). Like the classic stimuli [9], each image contained a three-dimensional object that was built of cubes connected face-to-face like a lego, with multiple arms and two free ends. To ensure stimulus diversity, each stimulus was constructed using either 10 or 13 identical cubes ($1m^3$), yielding objects with 3 and 4 connected arms respectively. The camera was positioned 25m away from the centroid of the object. For images in the training set, the direction of the camera was allowed to be completely random in 3D space. The test images comprised a subset of 10 objects, and the camera direction was varied systematically by choosing equally spaced angles along the the horizontal plane and the vertical plane, yielding 20 unique views per object.

### 2.2 Model

The model was comprised of two modules (Fig. 1). An autoencoder compressed the input to a low-dimensional latent space, and a recurrent neural network received a pair of latent embeddings and generated a time-varying decision variable as the output. Unlike models that learn to predict rotation angle with supervision [10], this model was not trained end-to-end. Instead, the modules were optimized with separate, general-purpose objectives as described below.

The autoencoder module was a feedforward network comprised of an encoder with three layers of width 128, 64, and 12 ReLu units, and a decoder with an inverted structure to reconstruct the input. The input dimensionality was 4096 and the autoencoder was trained with $L_2$ regularization to minimize the reconstruction loss (mean-squared error) across the training images. This module reduces the dimensionality of a 64x64 image by embedding it into a 12-dimensional latent space, $\mathbf{x}$.

The second module was a vanilla recurrent neural network (RNN) comprised of 200 hidden units with tanh non-linearity, two 12-dimensional input channels that received inputs $\mathbf{x}_1$ and $\mathbf{x}_2$, and a linear readout that generated a one-dimensional time-varying output, $y(t)$. The network was trained via back propagation through time, to integrate evidence for sameness at a rate that was proportional to the proximity (inverse of distance) between the inputs, until a bound of 1 was reached. The resulting target function for the output was $y(t) = \min(ct/|\mathbf{x}_1 - \mathbf{x}_2|, 1)$. The constant of proportionality, $c$, was adjusted such that the mean response time across the training set was 0.6 seconds.

### 2.3 Evaluation

Next, we analyzed the latent representation learned by the autoencoder on the set of 200 test images. We computed the distance between the latent embeddings of all pairs of test images containing the same object at different rotations, and quantified the relationship between distance and the angle of rotation by linear regression.

Finally, we computed the response time for each pair of latent embeddings tested above, by feeding them as inputs to the RNN module. The response time was taken to be the time $t$ at which the output $y(t)$ reached an amplitude of 0.95. This can be interpreted as the time at which the network is at least 95% confident that the two inputs correspond to the same object.

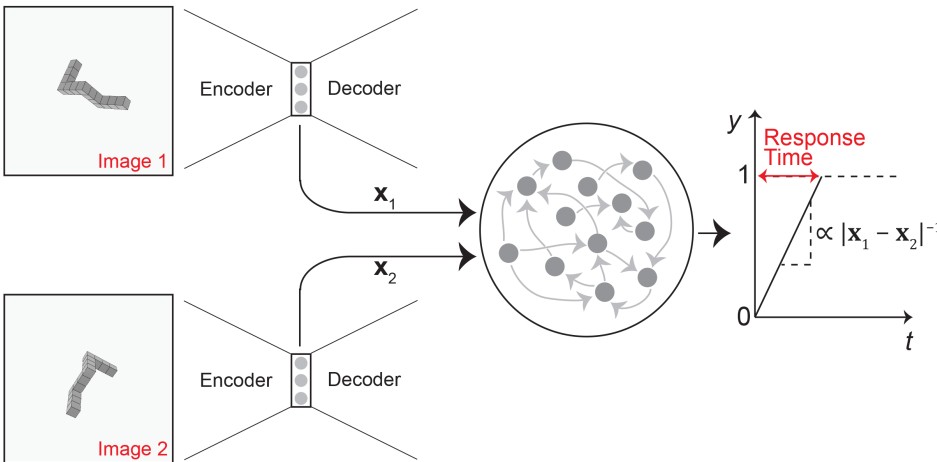

Figure 1: The model takes in a pair of images and first embeds them into a low-dimensional latent space. The compressed visual representations are then fed to a recurrent module which accumulates evidence for sameness at a rate that is proportional to the proximity of the inputs. Response is triggered when a bound (horizontal dashed line) is reached.

## 3 Results

We verified that the autoencoder produced near-perfect reconstructions of all images implying that the compression was nearly lossless ($R^2 = 0.98 \pm 0.01$; Fig. S2A,B in Appendix). We examined the distance between the latent representation of all image pairs, and found that the embeddings of images containing the same object were closer to each other than those that contained different objects (Mean distance $\pm$ SEM (a.u.) – same objects: $20.5 \pm 2.4$, different objects: $81.5 \pm 0.8$; Fig. S2C). This means that the low-dimensional visual representation learned by the autoencoder is informative about the sameness of objects. We tested whether the latent representations also contained fine-grained information about the angle of rotation. Consistent with our hypothesis, the distance between latent representation of test image pairs containing the same 3D object was significantly correlated with the magnitude of rotation of the objects in those images (Fig. 2A; Pearson's $r = 0.52$, $p = 1.7$e-3).

The recurrent network module showed a linear increase in response times as a function of the distance between inputs during training (Fig. S3 in Appendix), and this behavior generalized to the test set (Fig. 2B; Pearson's $r = 0.83$, $p < 1$e-8). Taken together, the two modules can explain the rotation-dependent increase in response times observed in spatial reasoning tasks (Fig. 2C).

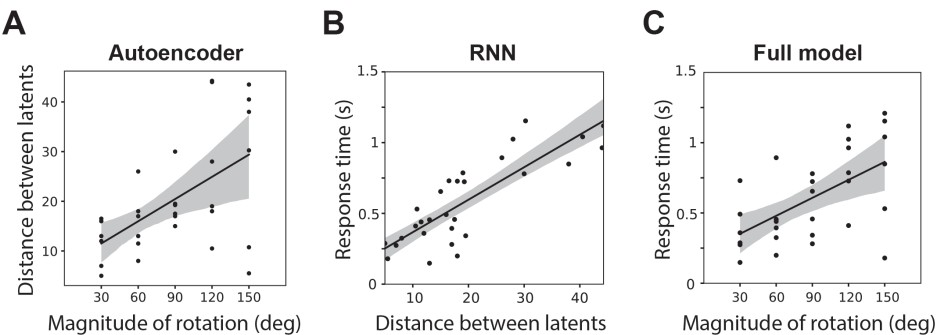

Figure 2: A. Distance between latent representations of image pairs as a function of the magnitude of rotation of the object. B. Response time of the RNN module as a function of the distance between inputs. C. Response time as a function of the magnitude of rotation. Shaded regions denote 95% confidence intervals.

# 4 Discussion

The similarity between representations learned by feedforward neural networks and visual cortex has been one of the most remarkable success stories linking human and machine intelligence [11]. It is generally believed that this similarity might not have implications for complex cognitive tasks which go significantly beyond the realm of image recognition [12, 13, 14]. Our findings show that this need not be the case. The geometry of representation of 3D objects learned by even a purely unsupervised mechanism, has sufficient structure to facilitate a rich, human-like pattern of response times in spatial reasoning. This is in alignment with recent studies that demonstrate that visual representations may be sufficient to perform physical inferences about the stability of objects without any physics engine [7, 8].

Our modeling approach is in the spirit of leveraging the representational capacity of feedforward networks in the service of flexible cognitive processes [15]. In particular, the autoencoder module extracts a low-dimensional visual representation in a task-agnostic manner, while the recurrent module implements a drift-diffusion process that sequentially samples from that representation until a judgement about sameness is made. In contrast, existing accounts of behavior in this task downplay the contribution of visual representations and instead appeal to a purely hypothetical cognitive process called mental rotation. Rational models of mental rotation have had to additionally grapple with the problem of how to determine the axis and the direction of rotation, and how much to rotate [16]. Mechanistic models of mental rotation attempt to achieve rotation-invariant pattern recognition by positing finely tuned attractor dynamics that drift toward a canonical view of the object stored in memory [17]. We suggest that models that allow visual representations to do some of the heavy-lifting can substantially reduce the cognitive demands of spatial reasoning. A dominant role for visual representation would also explain why human responses in this task are not cognitively penetrable [18].

We emphasize that although the proposed model does not feature an explicit mechanism for mental rotation, it does not falsify the mental rotation hypothesis. One way to accommodate a role for mental rotation in this framework is by adding feedback connections from the recurrent module to the bottleneck layer of the autoencoder. Such a modification would induce time-varying activity in that layer, allowing for the representation of one input image to evolve towards the other. We note that for dynamical representations to qualify as evidence for mental rotation, they must satisfy an important constraint: the intermediate representations should correspond to valid latent embeddings of the object that smoothly interpolates between the two 3D views. Neuroimaging studies have consistently found that the human posterior parietal cortex is activated during this task [19]. Whether this activity is a substrate for mental rotation or not would depend on the precise temporal dynamics.

The proposed model exhibits greater variability in response times than behavioral data reported in literature. Whether this discrepancy is due to the form of synthetic images used here is unclear but could be tested experimentally. In keeping with the large body of work on spatial reasoning, we tested our models using a simple set of 3D stimuli in which only the rotation angle was varied. Future extensions of this work should analyze how added sources of variability in object texture, lighting condition, and viewing distance interact with rotation angles to influence decision times, and test those predictions in humans. Another limitation of this model is that it does not consider the contribution of active sampling strategies using eye movements [20]. Directed eye movements have been shown to play a critical role in many cognitive tasks. Therefore, it would be important to extend this model by allowing visual representations to be influenced by spatial attention. Finally, the RNN was trained to linearly integrate the difference between inputs. While this has an appealing interpretation due to its correspondence with the drift-diffusion process, future studies must test whether alternative loss functions provide quantitatively better fits to human response times.

## Acknowledgments and Disclosure of Funding

The authors thank Lakshmi Govindarajan, Kohitij Kor, and other participants and teaching assistants of the Brains, Minds and Machines Summer school conducted at the Marine Biological Laboratory for several helpful and inspiring conversations and suggestions. This work was supported by the NSF Science and Technology Award to the Center for Brains, Minds and Machines (CBMM), and in part by the NSF NeuroNex Award DBI-1707398 and the Gatsby Charitable Foundation.

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

# Appendix

## .1 Method details

Code to generate the stimulus set, and to train and analyze networks can be found in this public repository: https://github.com/kaushik-l/mentalrotation.

When rendering the images using Blender, six point light sources were placed around the object at equal distances along the cardinal axes. The brightness and the location of these sources can be found in the code.

Images were transformed to grayscale and normalized to have a mean and standard deviation of 0.5 before training. The autoencoder was trained for 30,000 epochs with a batch size of 128, learning rate of 1e-3, and weight decay hyperparameter set to 1e-5. Training was carried out using 'Adam' optimizer. The decoder output was tanh transformed to keep the dynamic range between -1 to 1. We also optionally reduced the dimensionality of the bottleneck layer from 12 dimensions down to 3 dimensions by applying a linear transformation, and found no degradation in performance.

The time constant of the RNN hidden layer units was 100ms, and the network activity was updated in timesteps of 10ms. The RNN was first trained using pairs of simulated input vectors for 200,000 epochs and then retrained with the pair of latent embeddings from the training images for a further 10,000 epochs. The learning rate was set to 1e-3 and training was carried out using 'Adam' optimizer. Note that the response times of the RNN was set to be less than 1 second to keep training times manageable. It should be straightforward to obtain longer response times ( 5 seconds in human behavior) by increasing the training time or the network size.

Dataset generation (CPU time about 20 hours) and training of the autoencoder module (CPU time about 2 hours) were carried out on a MacBook Pro (M1 chip, 2020), while the RNN was trained using TPUs available on Google Colaboratory (compute time about 5 hours).

## .2 Supplementary Figures

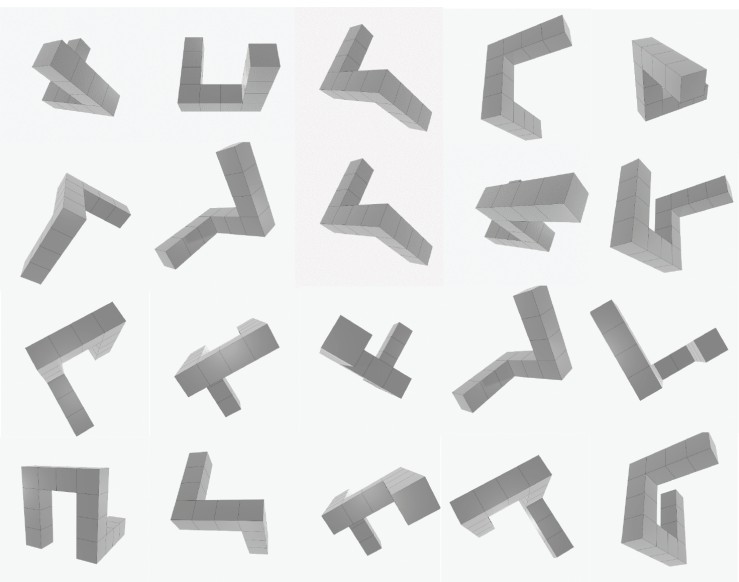

Figure S1: Close-up view of a selection of 20 synthetic images used in this study. Note that the experiments were conducted using images generated by positioning the camera further away from the object (see Fig. 1).

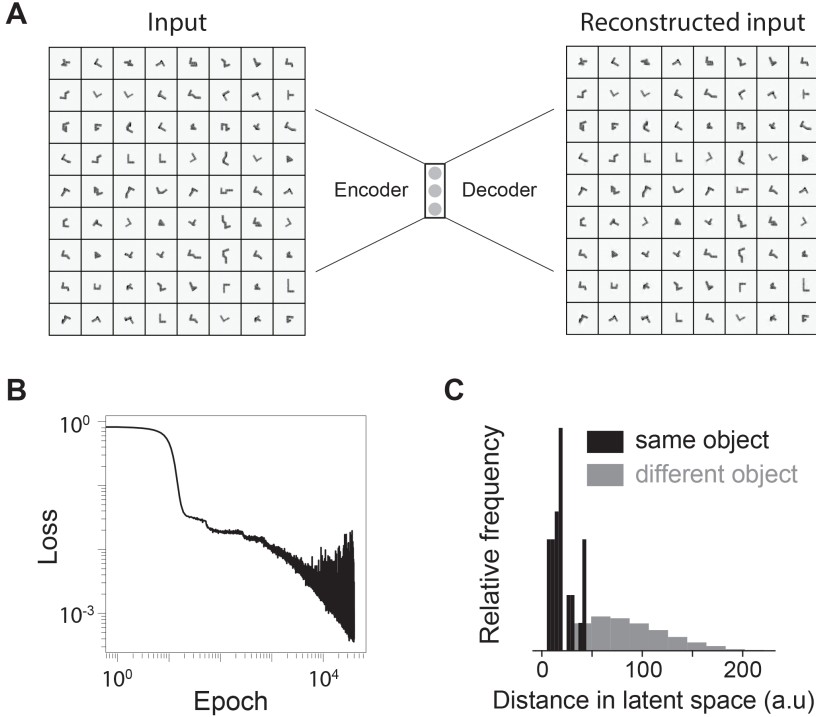

Figure S2: A. A subset of images (left) and their reconstructions (right). B. Reconstruction loss during training. C. Relative frequency of the distance (in latent space) between image pairs containing the same object (black) and different objects (gray).

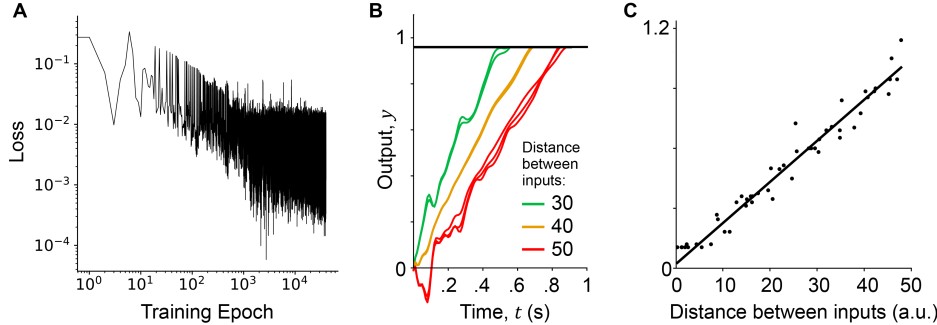

Figure S3: A. Loss during training of the RNN module. B. Output of the RNN for three different sets of inputs with different separations between them. Horizontal black line on top denotes the bound with respect to which response times are computed. C. Response times as a function of separation between inputs.

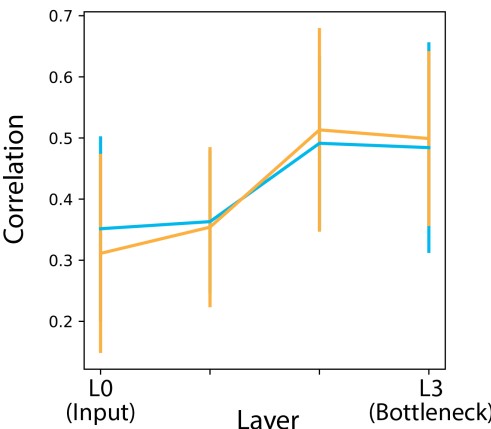

Figure S4: Correlation between model response times and angle of rotation, computed separately using the representation taken from each layer of the autoencoder. L0 corresponds to pixel space. Blue and orange lines denote results for two different axes of rotation.

