# OpenReview forum: "Unsupervised Representation Learning Facilitates Human-like Spatial Reasoning"
_NeurIPS.cc/2021/Workshop/SVRHM — SVRHM 2021 Poster_

### Official Review · Reviewer_D7C2 · 2021-10-29
**Novel modelling approach to highly debated psychological question**

**Rating:** 7
**Confidence:** 4

**Review:**

Shepard and Metzler found that the time taken by humans to judge whether two objects are the same, except for a rotation in 3D, depends on the relative degree of rotation. In cognitive science, there has been considerable debate as to whether the reason for this is because the task is performed by mentally rotating one of the objects, or if superficial features are compared, and things that are more rotated have fewer similar superficial features. The current project addresses this in a novel and interesting way, using an autoencoder to generate an embedding of the images and an RNN to compare the embeddings. The results are consistent with the feature account.

The work is generally clear, but it has some significant omissions. It is disingenuous to suggest that this is the first time that the mental rotation account has been challenged. In the introduction, it should be acknowledged that the presence or absence of mental rotation has been strongly debated. For example, see the work of Zenon Pylyshyn, https://link.springer.com/article/10.3758/BF03196930 and work that has cited this. Authors such as Stephen Kosslyn have countered Pylyshyn's position, and could also be cited where relevant. Previous work on mental rotation using networks should also be considered (e.g., https://www.researchgate.net/publication/221566471_Mental_Rotation_by_Neural_Network)

The model is clearly explained. Will the code be shared? This would make the submission more attractive.

An additional baseline would be valuable. The embedding is only valuable if it is providing something that is not straightforwardly present in the image. It would be helpful to compare examine the similarity of the images, to see if this alone is a function of rotation. More comprehensively, the image difference and the difference at various layers in the model could be examined. I presume the authors will hypothesize that the embedding at the bottleneck of the autoencoder will be the most related to rotation.

From Figure 2C it is apparent that there is wide within-angle variation predicted by the model. This could be tested in humans. The data should be acquired, or this idea flagged in the discussion.

The final paragraph of the discussion is rather a bombastic leap. I would suggest that the main value of this work is to show that network models can be useful in testing hypotheses from cognitive science, rather than pointing to a reimagining of how humans think.

MINOR POINTS
line 29: underlies -> underlie
line 56: delete that

---

### Official Review · Reviewer_4duZ · 2021-10-31
**A computational model that emulates human response times in mental rotation tasks**

**Rating:** 7
**Confidence:** 3

**Review:**

This paper proposes a computational model to investigate the neural mechanism by which humans assess the sameness of two objects. The authors generate a dataset of 3D objects and design a computational model that assesses the sameness of two objects. The model consists of two parts a convolutional autoencoder that learns a low-level representation of the object and a recurrent layer that integrates evidence for sameness at a rate inversely proportional to the distance of two objects in the latent space. The results show that the proposed model explains the increase in response times with the increase in the magnitude of rotation similar to human behavioral studies.

Overall the paper is well written and the text is easier to follow. The use of a recurrent layer to integrate evidence of sameness is an interesting approach to emulate responses times. The idea could possibly be extended to emulate the response times of several other human behavioral experiments.  I like the simplicity of the approach and the clear message this paper provides.

Here is a suggestion that I think might make the contribution stronger:
While the choice of the loss function for the recurrent layer makes sense, it might be a good idea to investigate some other choice of loss functions as baselines. One example would be to explore polynomial/exponential/logarithmic functions in recurrent layer loss function instead of linear one and then compare the relationship of response time with rotation angle. Then comparing different model response times to human response times using the same stimulus set used in this paper on the same task would be a great addition to reveal the neural mechanism of  how humans assess the sameness of two objects

---

### Official Review · Reviewer_rBjA · 2021-11-01
**An intreaguing hypothesis and a simple experiment**

**Rating:** 5
**Confidence:** 3

**Review:**

This paper models humans' similarity judgment of 3D objects under different rotations. The model consists of a feed forward module for visual perception, trained with an unsupervised objective and a recurrent module trained to compare latent representations. The proposed spatial reasoning account challenges the dominating hypothesis of mental simulation in the literature. Synthetic data was generated and used to test the proposed model.

## Strengths

- The paper challenges claims that similarity judgment in humans depends on a cognitive process. Since it can be possible to judge similarity using unsupervisedly learned features on a simple dataset, solving it shouldn't require an expensive cognitive process.
- The paper proposes a simple synthetic dataset for models of mental simulation.
- The paper is easy to read due to the simple structure and wording in the paragraphs.
- The authors provide model components and a detailed descriptions of the computational resources they used.
- Code is provided for reproducing the results.

## Weaknesses

major:

- The RNN is trained to follow response times that are linear with respect to the distance between representations. It's obvious that response times are linear with respect to distance between latents. Thus, S3 - B and C are obvious results and the result in figure 2 C is a direct result of figure 2 A.
- Throughout reading, it's unclear whether the RNN was trained to output a binary prediction (yes/no) of the sameness. If it was trained, no prediction accuracy of the recurrent model was reported.
- Even though a simple autoencoder is able to model simple object rotations, the autoencoder doesn't necessarily scale to natural objects and scenes due to their complexity. Autoencoders are not able to fully disentangle factors that generate a scene (such as rotation, object color .. etc). Since disentanglement is necessary for performing similarity judgment, autoencoders would probably fail at large scale image datasets.

minor:
- No model analysis was conducted (attractor analysis of the RNN, PCA or TSNE over the latent space as simple examples).
- No alternative models in the literature were tested.

---

### Decision · Program_Chairs · 2021-11-02

Accept (Poster)